# Methodology to Evaluate Economic Viability Plans and Digitalization Strategies in Private Social Education Centers

**Ricardo Francisco Reier Forradellas** [1,*], **Javier Jorge-Vázquez** [1], **Sergio Luis Náñez Alonso** [1] and **Ricardo Salazar Valdivia** [2]

1 DEKIS Research Group, Department of Economics, Catholic University of Avila, 05005 Avila, Spain; javier.jorge@ucavila.es (J.J.-V.); sergio.nanez@ucavila.es (S.L.N.A.)
2 El Carmelo School, 18006 Granada, Spain; ricardo@elcarmelogranada.es
* Correspondence: ricardo.reier@ucavila.es

**Abstract:** The Spanish educational system is characterized by the coexistence of three different models of production and provision of education: public, subsidized and private. Within the privately-owned centers not under the subsidized system, private schools of a social nature stand out. These schools, whose main source of financing comes from the fees paid by the students' families, must implement financial strategies that guarantee their economic viability and allow them to develop their educational project. In a highly competitive environment, the implementation of sound financial strategies and the development of educational innovation policies are critical to ensure their survival. In this context, this study analyzes a methodological proposal that can contribute to guide this strategic policy based on two fundamental pillars: the financial viability of the center and educational innovation through the application of new technologies and innovative teaching strategies. To this end, the case method has been used as the main methodology, obtaining results that considerably improve student satisfaction and that represent economic improvements of more than €100,000 per year. From these results it has been possible to identify different possible scenarios that can condition the financial viability of the educational center, the dropout rate and the academic performance of the students.

**Keywords:** digital transformation; financial feasibility; non-face-to-face learning; financial sustainability; learning platform; financial performance; savings generation; educational system; teaching innovation

## 1. Introduction

Education is one of the fundamental pillars of a country that determines the future development of society. Improving the system has a direct benefit not only on educational results, but also on the resources available to the public coffers. Each school failure in the USA costs $260,000 to the public coffers (extrapolated to the 16% school failure rate in the country as a whole) [1]. In Spain, the data are more alarming, according to the report on the private and fiscal profitability of education in Spain by the Observatory on Human Capital in Spain of BBVA Research, which indicates that the cost of school failure represents 60% of the direct expenditure of the public sector [2].

On the other hand, in recent years it is an unquestionable fact that the development of new digital technologies has led to a transformation of our society. This new technological paradigm has opened up a wide range of possibilities unsuspected in the field of education [3].

At present there are several authors (Alonso, Cebrei, Dussel and Quevedo, Pedreño, García Aretio) who have pointed out the positive influence of new information and communication technologies in improving teaching–learning processes [4–9].

Returning to the specific case of the educational system in Spain, the evolution has been positive thanks, to a large extent, to the investment that has been made [10]. The following

figure (Figure 1) shows the evolution of the level of education in Spain in comparison with Organization for Economic Cooperation and Development (OECD) member countries.

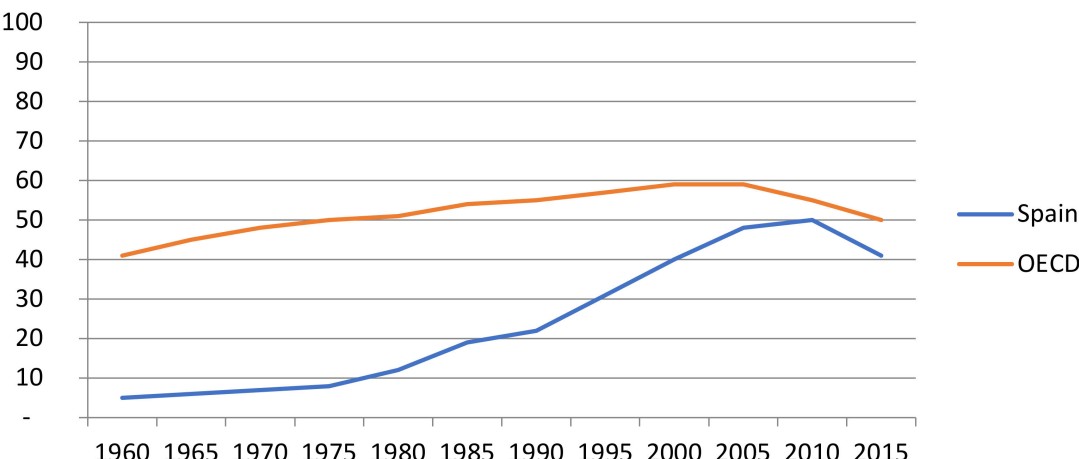

**Figure 1.** Percentage of the population over 25 years of age with secondary but not university education. Spain, Organization for Economic Cooperation and Development (OECD) (1960–2015). Source: own elaboration based on De la Fuente, A.; Domenech R. [10].

As can be seen, Spain still does not reach the OECD educational average and is still far from the highest educational positions. However, Figure 2 shows a positive evolution in educational results that is linked to an increase in investment.

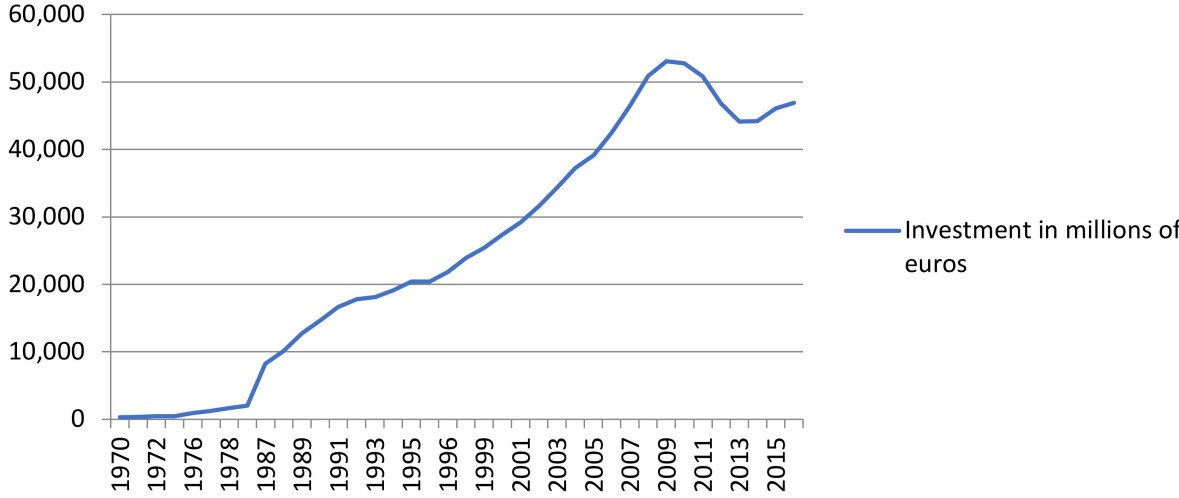

**Figure 2.** Evolution of investment in education in Spain between 1970 to 2015. Source: own elaboration based on Jurado Sanchez, M. [11].

These data clearly indicate how an investment in education has results and is much more profitable than trying to placate the consequences of school failure [11].

At present, education in Spain is governed by public, subsidized and private centers whose economic management is significant in terms of the rate of success per investment. Traditionally, private and subsidized education manages economic resources more efficiently and with better results [12]. It is true that social biases have an influence, but evidence shows that the school failure rate in public educational centers is usually double and in some cases triple that in private education, whether subsidized or exclusively private [13].

In Spain there are mainly two models of private centers; those that come from a purely business philosophy and those that have a more traditional social character, inherited from



the functions they performed in the past. According to information gathered by the Spanish National Statistics Institute (INE), the former tends to choose a specific market based on a client profile with a high purchasing power and the ability to pay around €5000 on average, which usually accounts for 18.5% of incomes of over €3000 per month [14]. As an example of the latter, Christian religious centers have a historical origin and a social character that normally do not seek a business profit that makes them compete economically, but rather, due to the political interests of the country, they have provided educational services in order to respond to the social and business demand [15].

At present, the public agreement represents a clear saving for the public coffers, which cannot assume the costs that would be involved in providing service to all students of compulsory school age. In addition, these centers have a lower maintenance cost than public ones [16]. This is the reason why there has been a significant increase in the number of private centers acquiring the agreement in recent years [17], as can be seen in Figure 3.

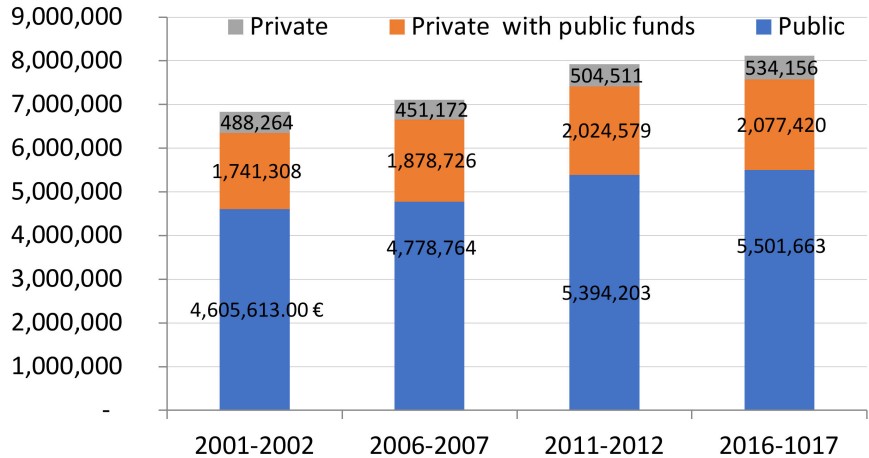

**Figure 3.** Compulsory schooling by sector 2000–2017. Source: own elaboration based on Ministry of Education, 2018.

As can be seen in the figure, the total number of students has been increasing in all categories, going from a total of 6,835,185 in the 2001–2002 academic year to 8,113,239 in 2016–2017.

However, the reality is that private schools cannot compete on price with public education or with subsidized schools, which has forced private schools to follow different management strategies: those that target a client with a high purchasing power by offering exclusivity, better quality facilities and educational quality, and those that, due to their business paradigm, seek to provide an education in values in line with the identity of the center and economically demand the minimum for its operation. This last specific case is the focus of this paper.

Part of the sector of the population that would like to opt for non-subsidized private education is conditioned by a specific socioeconomic reality that limits this possibility. The figures indicate that 45.6% of workers are paid less than €18,345 per year and that 39.3% of the salary of Spanish workers went to the payment of taxes and social security contributions in 2017 [18]. This reality also limits access to all types of non-public centers.

This socioeconomic reality in the specific Spanish case raises one of the key questions for the model of private education with a social initiative: what fee must it have for the center to be economically viable and able to attract students? This situation makes it difficult for private centers with a social business paradigm to survive in a competitive environment and to avoid being "phagocytized" by public and subsidized centers due to their price leadership.

In order to provide a solution to this reality, it is necessary, from a practical point of view, to make a proposal to improve the current situation according to the general and

particular contextualization of this model of educational centers, from a strategic, financial and operational point of view.

## 2. Literature Review

When talking about different educational models, the first distinction that is often made refers to the concepts of public and private. However, even this conception of terms can lead to academic misunderstandings.

- The first distinction between public and private with regard to the concept of education is usually made on the basis of the origin of the school's funding. Thus, we speak of a public school when it is financed by some kind of public administration, and we refer to a private school when it is financed mainly through fees paid by the pupils' parents.
- A second distinction, also very common, distinguishes the educational model according to the type of management of each educational institution. That is to say, regardless of funding, this distinction focuses on whether the management is private or public.

One could even go further and make distinctions between public and private depending on the nature of the goods produced. In this sense, authors such as Levin, Noddings and Silveira define public education as education that produces public goods, and private education as education that produces goods of a private nature [19–21].

Generally speaking, as authors such as de Pablos point out, education has historically been considered one of the most important factors in the socio–economic development of a country and, traditionally, it has been the state that has been responsible for promoting and protecting the formation of human capital [22]. Temprano and Villanueva also link public spending on education as one of the traditional objectives of the state to universalize a certain level of social welfare [23]. In fact, recent studies such as those by Cardenas Zambrano or those published by institutions such as the International Monetary Fund (IMF) show how public spending on education has had a greater influence than public spending on health in reducing social inequality [24,25]. In this way, it can be concluded that, in general terms, there are a number of services provided by the public sector in the form of services. These services will vary from state to state, depending on the form of provision, but there is a certain generality in the case of education and health care [26].

Returning to the classification regarding the origin of funding for the distinction between public and private, one could define as public those educational institutions which are directly financed by public funds and private as those which are financed by private funds. These private funds will come mainly from the pupils' fees and, to a lesser extent, from other concepts such as corporate donations, funds from religious institutions, etc. This distinction has traditionally been taken for granted in the academic field by authors such as Hanushek and Woessmann who point out that this type of distinction between public and private is relevant because there are significant differences between one model and another [27]. Similarly, authors such as Pasaran have highlighted the educational differences between public and private models in terms of the results obtained by students [28].

With regards to the management model of each school, it is common to define privately managed schools as private schools and those that depend directly on the administration as public schools. In this sense, authors such as Glenn, Goldstein and Vandenbergue link the public education model to the administrations' intention to guarantee a centralized, homogeneous and accessible educational model for all citizens [29–31].

Particular importance has been given to these two aspects in relation to school ownership (especially the one referring to the origin of funds) as this is the basis of this paper. However, these distinctions are not categorical concepts and there is currently a great deal of complexity when talking about pure forms of funding. The following pictures show (Figure 4), as an example, the distribution of the different educational institutions in some European countries.

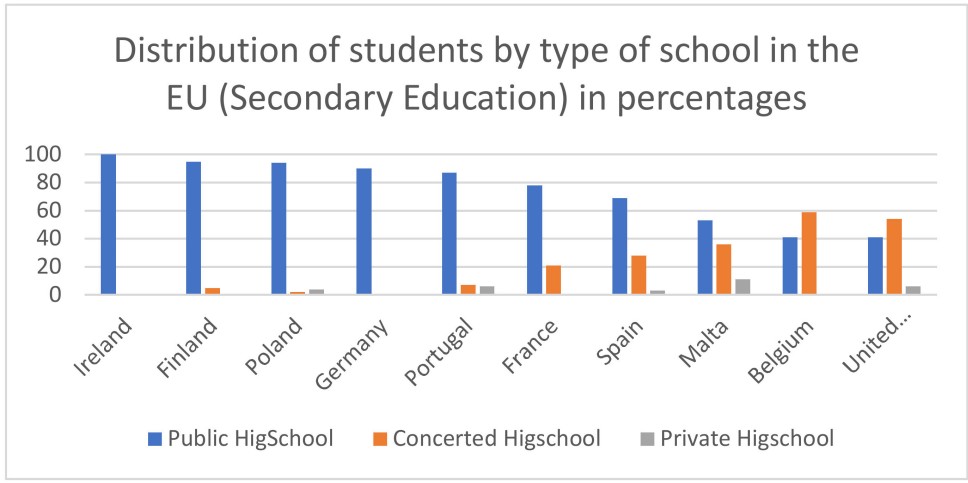

**Figure 4.** Compulsory schooling by sector, 2000–2017. Source: own elaboration based on Eurostat, 2018.

However, this distinction is not entirely clear. Today, most education systems have resources from both the public and private spheres. It is common to find educational models in which tuition fees come from private sources but are combined with state subsidies for the purchase of books, computer equipment or uniforms. In other cases, particularly in the Anglo-Saxon context, the basic public budget is supplemented by a series of donations from the private sector. It is even common for publicly owned school buildings to be taken over by private or religious organizations for educational purposes. Thus, the public–private relationship in education is relatively common. In fact, when speaking in terms of public or private, what is usually referred to is the majority percentage of funds received, and the management model. In this way, looking only at the level of funding, and analyzing different countries as examples:

- In countries such as the Netherlands and Germany, private pre-university education is almost non-existent, and education is basically public. Private institutions occur mainly in the field of higher education.
- In Denmark, the education system offers both public and private education, but private education would be up to 80% publicly funded.
- In countries such as Italy, there are both public and private schools. The latter (private schools) are, as a rule, also supported by public funds.
- In Austria, public and private schools also exist, although, as in the case of Italy, these private schools are usually also publicly funded.
- In the United Kingdom the model is also mixed, with private schools usually having a certain number of subsidized places for the poorer population.
- In countries such as the United States, education is highly decentralized and there are significant differences between states. However, private schools often have some form of public funding.

In this way, mixed schemes are becoming more and more frequent. These hybrid education systems have already been extensively studied in the academic field by authors such as Bellei, Orellana, Verger and Bonal, who have focused their studies on the relations and definition of responsibilities between the public and private sectors in the field of education. Their studies focus on the relationship and definition of responsibilities between the public and private sectors in the field of education. As already indicated, the existence of mixed educational models implies the existence of several alternatives in the privatization process [32,33]. The authors Verger, Zancajo and Fontdevilla speak of exogenous and endogenous privatization models [34].

- Exogenous privatization. This model would consist of providing facilities to private organizations to offer services in the education sector. These facilities could be provided

through public subsidies to private centers, tax incentives or policies to liberalize the education sector.

- Endogenous privatization. This model would consist of the application of "market rules" in education. In this way, concepts such as competition between schools, free choice for parents or incentives based on good educational results would be introduced. Perhaps the clearest form of this type of model is the system of school vouchers, which is widely used, especially in countries such as Chile and northern Europe.

The following table (Table 1) attempts to define globally and, simply as an example, the different paths of the two processes mentioned above.

**Table 1.** Global co-educational models. Source: own elaboration.

| Model | Countries | Changes Made |
|---|---|---|
| 1. Privatization of education as part of structural state reform | United Kingdom, Chile | Structural changes. Private actors in the education system. |
| 2. Education privatization as incremental reform | Uited States, Canada, Colombia | Gradual changes. Emergence of "charter schools", private management of publicly owned schools. |
| 3. Via Nordic | Scandinavian countries (Norway, Sweden, Finland) | Reformulation of the classic welfare state to modernize forms of provision and give citizens more choice. |
| 4. Historic public–private partnerships | Netherlands, Spain, Belgium | The state finances part of private education in exchange for a common regulation in terms of content, regulation, admission procedures, etc. |
| 5. Privatization "by default" | Low-income countries | Common in countries of sub-Saharan Africa, South Asia, Peru, etc. Appears in the LFPS (Low Fee Private Schools) |
| 6. Privatization via the "natural disaster" route | Countries affected by conflict and natural disasters | Sense of emergency due to special situations. Cases in El Salvador, Uganda, Guatemala, Haiti, Iraq, etc. |

As can be seen in this table above, as in Figure 1, Spain has one of the highest rates of students in private schools in the OECD. The implementation of an education system based on public subsidies to private schools (the so-called "concerted" schools in the case of Spain) took place between the end of the 1970s and the beginning of the 1980s. Due to the peculiarities of the Spanish case, the process of expansion of primary and secondary education did not take place after the Second World War as in the rest of Spain's neighboring countries. This process took place in Spain after the fall of Franco's regime and the first democratic elections in 1977 and the approval of the Constitution in 1978. Until then, as authors such as Bonal point out, the public authorities in Spain had an almost subsidiary presence in the field of education, having delegated these responsibilities mainly to the Catholic church [35]. In this way, authors such as Olmedo point out that through the so-called "1978 school pact", an education system was developed in which the state administration, in addition to public and directly private schools, was complemented by a series of private schools financed with public money [36].

Section 5 will analyze the different academic contributions to this educational model described for the case of Spain.

## 3. Materials and Methods

### 3.1. Materials

A first attempt to approximate the economic and financial reality that characterizes privately managed educational models requires the identification and selection of a diverse set of variables: average expenditure per student; distribution of the educational market between public and private education models; distribution of expenditure; and student profile according to purchasing power [37]. These same variables are defined by the Organization for Economic Cooperation and Development [38].

The main source of information used in this study comes from the official statistics published by the Spanish National Statistical Institute INE [14]. The figure below shows the average annual expenditure per student according to educational level and type of education. With this data, it is possible to establish a representative average cost of what parents are willing to pay annually for their children's education according to educational level, as can be seen in Figure 5.

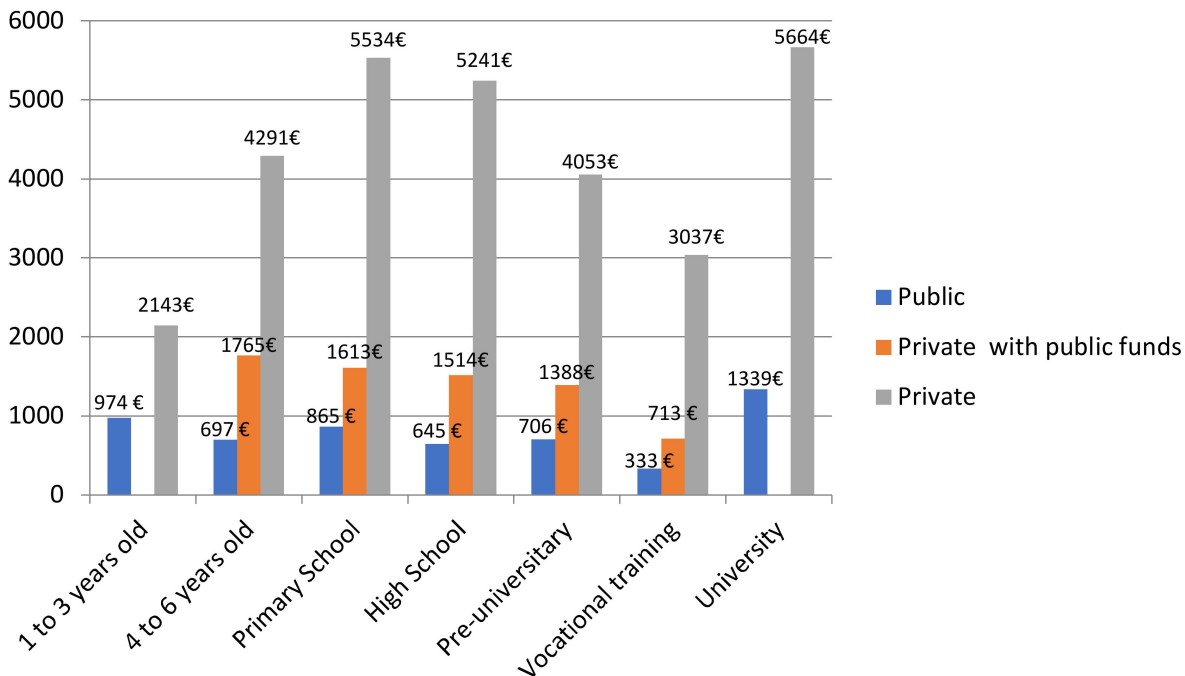

**Figure 5.** Average annual expenditure per student by educational level and school ownership in €, 2017. Source: own elaboration based on the Spanish National Statistics Institute INE 2017.

Based on the previous figure, it would be possible to estimate annual family spending per student according to the level of studies and the educational center model. However, in addition to the level of affordable expenditure, it is essential to know the percentage of the population that would be susceptible to attending a given center model. For this purpose, the distribution between income and population density are key factors that will help to make a reliable estimate [39]. The following tables allow us to relate the percentage of the population distributed among public, private and subsidized education according to the user's purchasing power and population density.

Table 2 refers to the general distribution of the type of educational center chosen according to income level.

**Table 2.** Distribution of formal education students by level of net monthly household income by classroom ownership, 2019.

| | | Ownership of the Classroom | | |
|---|---|---|---|---|
| **Net Monthly Income** | **Total** | **Public** | **Private Subsidized** | **Private Without Subsidies** |
| To €1499 | 100% | 80.90% | 14.30% | 4.70% |
| From €1500 to 2999 | 100% | 75.10% | 17.60% | 7.30% |
| €3000 or more | 100% | 57.50% | 24.00% | 18.50% |

Source: Own elaboration based on INE 2019.

Similarly, to complement the previous table, it is necessary to know the type of educational center chosen according to the students' area of residence based on population density. This information is shown in Table 3.

**Table 3.** Distribution of formal education students according to population density by classroom ownership. Academic year 2018/19.

| Population Density | Classroom Ownership | All Levels | Primary and ESO |
|---|---|---|---|
| **Densely populated area** | Total | 100 | 100 |
| | Public | 62 | 57.2 |
| | Private subsidized | 23.8 | 36.9 |
| | Private without subsidies | 14.2 | 5.9 |
| **Intermediately populated area** | Total | 100 | 100 |
| | Public | 71.7 | 71.8 |
| | Private subsidized | 15.4 | 23.5 |
| | Private without subsidies | 12.9 | 4.7 |
| **Lowly populated area** | Total | 100 | 100 |
| | Public | 79.1 | 84.2 |
| | Private subsidized | 10.3 | 13.3 |
| | Private without subsidies | 10.6 | 2.6 |

Fuente: source: own elaboration based on INE 2019.

A detailed analysis of educational spending necessarily requires that it be broken down into different items. In this breakdown, it is possible to identify different expenditure items linked to the supply of different goods and services: classes, extracurricular activities, canteen, school supplies, among others. It is important to break down and take these items into account when comparing educational centers since they can also influence the choice of one center or another [40]. The following figures (Figures 6–9) shows the different components of educational expenditure by center ownership and according to the different educational stages: primary, first and second cycle pre-school, ESO (Obligatory Higher Education) and Baccalaureate, based on data from the Spanish National Institute of Statistics [14].

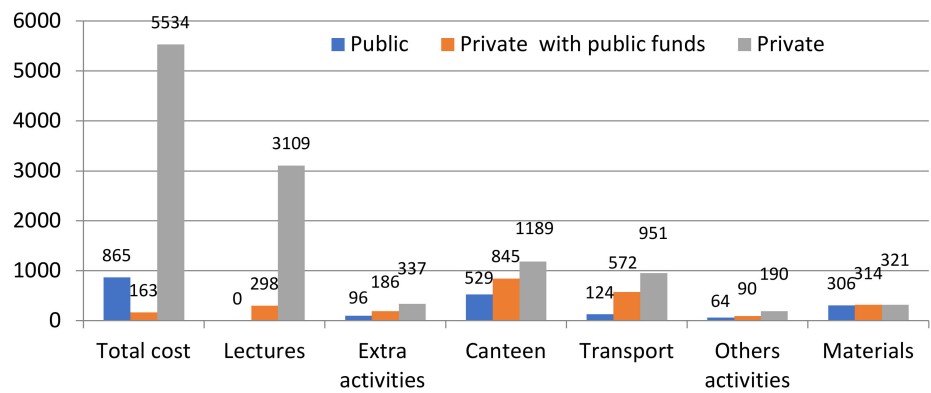

**Figure 6.** Main components of spending on primary education, in €. Spain 2017. Source: own elaboration based on INE.

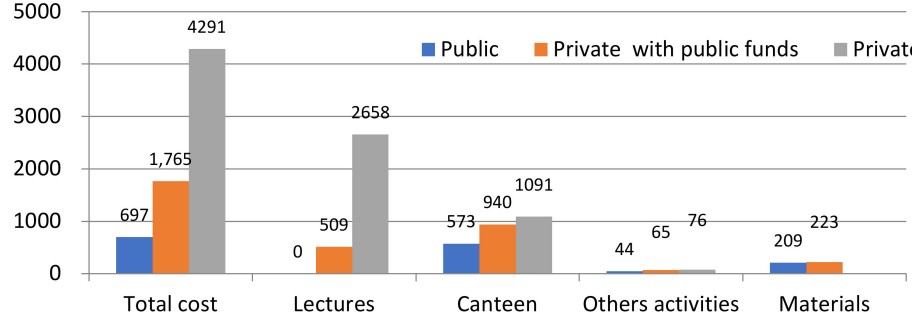

**Figure 7.** Main components of spendings on early childhood education, in €. Spain 2017. Source: own elaboration based on INE.

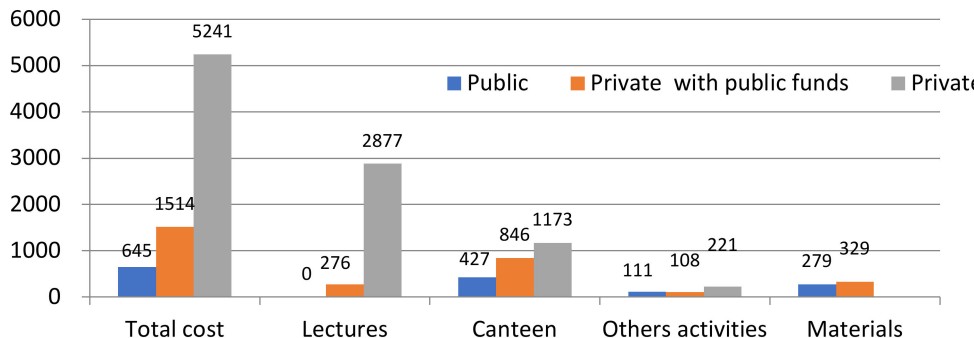

**Figure 8.** Main components of spending on obligatory higher education (ESO), in €. Spain 2017. Source: own elaboration based on INE.

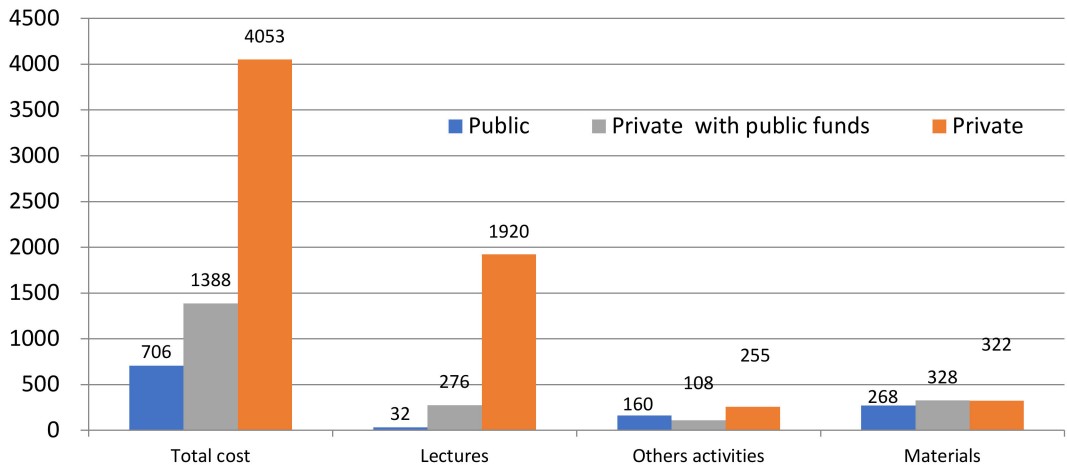

**Figure 9.** Main components of spending on pre-universitary studies, in €. Spain 2017. Source: own elaboration based on INE.

Based on these general data, the methodology that will determine the optimal management model for this educational center model will be developed.

### 3.2. Methodology

As mentioned in Section 1 and in Section 2, this work does not aim to assess the suitability of an exclusively public or mixed educational model, but rather to seek a viable alternative for a specific model of educational center that will subsequently allow the results to be replicated in similar centers. This work, as will be demonstrated in Section 6, aims to make this type of center financially viable from the point of view of sustainability, quality and educational innovation. In this way, the work will contribute to maintaining an educational model system such as the current one; in Section 5, the contributions made by various authors regarding the suitability and problems posed by the existing educational system in Spain will be analyzed from an academic point of view.

As already indicated, the methodology used will be the case study. Case study research offers the opportunity to go beyond the specific understanding of an isolated case, allowing the case to be used as a reference element to interpret and apply the results obtained more broadly [41,42]. As will be seen below, this type of methodology is common in the field of social sciences and, more specifically, in the field of education as it allows similar realities to be assessed from a homogeneous perspective [43].

Although it is true that the existing academic information on the use of the case study method in scientific research is scarce [44], it is a valuable tool that allows the empirical recording of the behavior of the actors involved in the phenomenon under study, increasing the results that would be obtained from research based solely on quantitative methods through tools such as surveys or questionnaires [45]. Through this methodology, it is

possible to obtain the data to be analyzed from a variety of sources, both qualitative and quantitative, being an essential form of research in the social sciences, as well as in specific areas such as education. Thus, although the case study was traditionally considered appropriate only for exploratory research, some of the works with this methodology that have had the greatest impact have been both descriptive and explanatory [46].

With regards to their purpose, research carried out using the case study method can be descriptive (-the aim is to identify and describe the different factors that influence the phenomenon under study) and exploratory (the aim is to achieve a rapprochement between the theories included in the theoretical framework and the reality under study) [47].

Following the aforementioned methodology based on simulation and case analysis when applying the concepts developed, the study will be carried out on a concrete model of an educational center, from which it will be possible to contextualize and develop a global quantitative and qualitative model. The methodological approach, therefore, will be oriented towards the development of the procedures and elements necessary for the use of the case study method as a methodological tool for scientific research. This methodological process of scientific research will include the inductive-hypothetical-deductive phases [48] that will allow the final objective to be achieved: to demonstrate key characteristics such as the value, benefit and practical usefulness of the method and to link its scientific validity associated with quantitative methods.

The case method will seek a generic model of an educational center on which to subsequently apply the methodology developed. For the observation and description of the phenomenon to be studied, a private religious center (social character) located in a specific locality of the Spanish territory will be chosen. A specific model center will be used that meets the generic conditions defined by the Ministry of Education in terms of average population density, volume of income of private religious centers and level of competence. To support these data, information will also be collected from both public institutions (the European Commission) and the academic world [49–51]. Logically, it could be extrapolated to any other center in a similar situation and with similar characteristics. As indicated, a standard center model has been chosen, including a specific location, in order to be able to carry out the study according to sociodemographic characteristics. For other specific cases, it would only be necessary to adapt the reality of each center to the model and methodology to be proposed. Thus, from a specific case study, it will be possible to develop a methodology applicable to the overall reality of the object of study, methods and ideas to be developed with the reality of the object of study [52].

Continuing with the definition of the case study as a research strategy aimed at understanding the dynamics present in singular contexts [53], we will consider a concrete model of an educational center located in the municipality of Granada, within the Autonomous Community of Andalusia, Spain. A specific locality must be taken since in the methodological development (when designing the surveys) it will be necessary to compare the data obtained with those provided by the Ministry of Education, broken down at the territorial level. Thus, an area of medium-high population density with a typical per capita income of €15,764 will be considered. In the same study, it will be estimated that within a radius of 3 kilometers, there are two public schools and one subsidized school in competition [54].

Based on the previous case model, the generic activities that generate income for any educational center similar to the one proposed are used as a starting point:

- Teaching
- Rental of sports courts for extracurricular activities
- Renting of the assembly hall
- School canteen.
- Similarly, the cost-generating activities would be the following:
- Cost of teaching staff
- Cost of maintenance and secretarial staff
- Cost of maintenance of the center's facilities
- Management and administration costs

- Costs of consumables for teaching
- Computer maintenance costs
- Cost of the school canteen.

Based on these two income and cost indicators, and based on the information gathered in the introduction, the main objective will be to develop a strategic plan to improve the situation of this type of educational center based on the analysis of the proposed case, and to eliminate the economic problems derived from financial and operational leverage [55]. The specific objectives are as follows:

1. SO1: Conduct a strategic analysis.
2. SO2: Propose a strategic and financial improvement proposal.
3. SO3: Establish a strategy feasibility and planning plan.

The final feasibility of the financial strategies that may be derived from the methodology used will be subject to final assessment through surveys to parents and students (parents and students essentially). The inclusion of surveys allows the verification of the results and constitutes an adequate complement to the scientific methodology developed in the work, being considered as a research technique [56]. As various authors maintain, the use of surveys has become one of the most widespread research techniques, going from being considered a mere technical instrument for data collection to a research method [57]. The methodological combination of the case study method with the use of surveys as a scientific tool will make it possible to define the methodological approach of the present work [58]. The survey models to be used are available in the appendix.

### 3.2.1. Development of a Strategic Analysis (SO1)

Based on the quantitative variables collected in Section 3.1, this information is complemented with the data collected in Table 4, which are necessary for the proposed methodological development.

**Table 4.** Sociodemographic data for the case study.

| Generic population of the locality to be considered | | 47,000 habitants | |
|---|---|---|---|
| Population between 4 and 18 years old (16%) | | 7533 habitants | |
| % of population with income up to €1499 net per month | 4.70% | % of population with income between €1500 and 2999 per month | 7.30% |
| Potential students | 354 | Potential students | 550 |
| % of population with income up to €1499 net per month in densely populated areas | 12.80% | % of population with income between €1500 and 2999 per month in densely populated areas | 14.70% |
| Potential students | 964 | Potential students | 1107 |

Fuente: Source: own elaboration based on INE.

From Table 4, we obtain a range of possible students from 354 in the worst-case scenario to 1107 in the best-case scenario. A center with a capacity for 700 students in optimal conditions will be estimated. This number of 700 places, in optimal conditions, is also determined by the Ministry of Education as the average number of places offered in this type of educational center [54]. It is possible that there may be more students per classroom, but the quality of teaching could suffer, so a maximum of 25 students per classroom in primary and secondary education, 20 in high school and 15 in early childhood education [59] is considered.

In the most unfavorable situation (as seen in Table 5), we would have 354 students with an average total annual expenditure per student of €1,728,875.31 and an average annual expenditure per student of €4883.83. In the most favorable situation, there would be 700 students with a total expenditure of €3,418,680.00.

**Table 5.** Average spending per student for a center with the dimensions of the case.

| Educational Stage | Average Spending Per Student and Per Course | No. of Students | Total Average Spending |
|---|---|---|---|
| EARLY CHILDHOOD 1. CYCLE | €2143.00 | 60 | €128,580.00 |
| EARLY CHILDHOOD 1. CYCLE | €4291.00 | 60 | €257,460.00 |
| PRIMARY | €5534.00 | 300 | €1,660,200.00 |
| ESO (COMPULSORY SECONDARY EDUCATION) | €5241.00 | 200 | €1,048,200.00 |
| PRE UNIVERSITARY (BACCALAUREATE) | €4053.00 | 80 | €324,240.00 |
| | | 700 | €3,418,680.00 |
| | | 354 | €1,728,875.31 |

Source: Own elaboration.

Assuming an average occupancy rate of around 95% (standard figure) [54], the center used as a case study would have an occupancy of 664 students. Assuming an annual tuition expenditure of €3000 per student (also a standard figure) [54], the annual income for the center would amount to €1,992,000.00. However, the cost of personnel alone for this number of students is €1,778,000.00 of the €2,006,900.00 that it costs to maintain the center annually with all its expenses (all of this is shown in Table 6).

**Table 6.** Cost of the case study period.

| | Indirect Costs | Direct Costs | Overhead Costs |
|---|---|---|---|
| Workers' salaries | | €1,778,000.00 | |
| Center maintenance costs | €100,000.00 | | |
| Expenditures on scholarships to needy/merit scholarships | | | €18,000.00 |
| Expenditures on school supplies | €99,600.00 | | |
| Cost of center activities | €2500.00 | | |
| Administration and management costs | | | €8800.00 |

Source: own elaboration.

Based on the above data, an analytical income statement has been prepared, which is shown in Table 7.

**Table 7.** Analytical income statement.

| | |
|---|---|
| Sales Revenue | €1,992,000.00 |
| Cost of Teaching Activities | €1,980,100.00 |
| Cost of Scholarships | €18,000.00 |
| Direct Result | €−6100.00 |
| Administration Cost | €8800.00 |
| Result for the Period | €−14,900.00 |

Source: Own elaboration.

The data collected in Tables 6 and 7 are based on works related to standard cost structures for models of non-subsidized private education centers, taking into account the number of students considered in the case study [38,60,61]. Therefore, we can establish that the bases on which the strategy should be based are those of a market ranging from 354 to 700 potential students with an average outlay in school expenses of €4888.83 (calculated from the average total expenditure divided by the number of students, based on the data in Table 4). In the case of starting with a standard figure of €3000 of expenditure, it is necessary to find a formula to obtain the remaining €1883.83, in order to improve the profitability of the proposed educational center model.

With this starting situation, the best possible strategy must be sought in order to guarantee the proposed objectives.

3.2.2. To Propose a Strategic and Financial Improvement Proposal (SO2)

When applying innovative methodologies in education, it is important to take the implementation of online content as a starting point. According to the results of the World Summit for Innovation in Education, 43% of experts believe that online content will be the main source of knowledge [62]. Similarly, the fact of adapting face-to-face/online secondary education makes it possible to increase the number of students and to opt for a more technical training model focused on vocational training profiles [63]. The application of new educational technologies will also increase the supply of extracurricular, leisure or social activities for the center, with a consequent increase in revenue [64]. This strategic model with more innovative educational projects is currently absorbing almost three quarters of the growth of the sector, with a student body that has grown by 53% [65]. On the other hand, according to the latest data from the Ministry of Education, slightly more than 122,000 students are studying for their baccalaureate and vocational training without attending classes in person [54].

Given the previous analysis, the most favorable option is the adoption of new technologies that allow the digital transformation of the educational offer: the incorporation of a digital teaching platform as an innovative and differentiating method in addition to opening the option of online studies with the intention of increasing the market range and therefore maintaining a stable number of guaranteed students. This method, even with the proposed methodological approach, including surveys, has already been addressed in other similar research relating teaching innovation and digitization [66].

The digitization of the classes will be done through a Moodle platform created and managed by each teacher. The use of the Moodle platform as a teaching tool for academic improvement, as well as its positive impact on the teaching–learning process, has already been discussed in many scientific fields at both pre-university and university levels [67]. There are also numerous studies that relate the use of virtual platforms with the improvement of school performance [68]. The administration itself (in this specific case, the Junta de Andalucía, the Autonomous Community, where the model educational center is located) has highlighted the advantages of applying these types of educational platforms (Moodle in this case) in the field of education, not only for distance models [69]. In this way, each teacher has the freedom to carry out his classes and methodologies based on this new tool. The aspect of server creation will be managed by the Technology and IT department. The center, from the management of the Direction, will be in charge of enabling the center to use these platforms for the online modality. As an added value to the incorporation of this platform, the application of virtual simulation systems will be proposed as innovative teaching systems so that the digitalization and innovation process is perceived as such by the students and their parents, allowing the center to apply an increase in the fee to be paid. The use of these interactive simulators accelerates the learning process and helps to improve its quality [70]. Many of them are free of charge and have academic papers demonstrating their importance in the educational process [71–73]. As has been pointed out, the implementation process will start with the free options in terms of virtual simulators available to any educational center (PhET, EduMedia, KDE, etc.), with teachers combining training on the Moodle platform with that of the simulators themselves.

This improvement in methodological and technological innovation makes it possible to justify an increase in the fee payable by students as a result of an increase in the value perceived by clients with respect to the quality of the services offered by the center.

If we consider the analyses already carried out, there would be around 354 potential students in the most unfavorable situation and 1107 students in the most favorable.

In our case analysis, as has already been mentioned, we propose a 95% occupancy rate with a unit income of €3000, corresponding to the fee per student. A substantial increase in the price could mean a loss of customers that could even be amortized with the increase in

the fee payable by the students and with the reduction in the number of staff resulting from a new online modality. However, this strategy does not usually fit into the social model of companies linked to these educational center models. Therefore, the initial proposal will be to increase the annual price per customer by €300. This amount has been established as it would correspond exactly to the price of one monthly payment; as mentioned, for this type of center, the average cost would be €3000 per year in 10 monthly payments, i.e., €300 per month.

The fact of offering an online modality may lead to an increase in the number of students, some of them even of different age segments and profiles than traditional students, especially in the Baccalaureate educational cycle, which could mitigate the possible loss of customers as a result of the 10% price increase.

Therefore, based on the reasons stated, the value proposition will consist of the integration of a digital platform where the contents, activities, evaluations, etc., of each course can be found, together with the application of virtual simulation systems to give the option to all students, both online and on-site, to make their studies more flexible. This aspect gives a differentiating value for obtaining income that will allow the improvement of the financial situation of this type of centers.

The key activity will be the combination of face-to-face and online digital teaching. This versatility is the key to the proposal, as it represents an innovative and technological improvement adapted to current needs and demanded by society. The global strategy in the financial area is shown in Figure 10, based on the financial data already referred to [54,60,61].

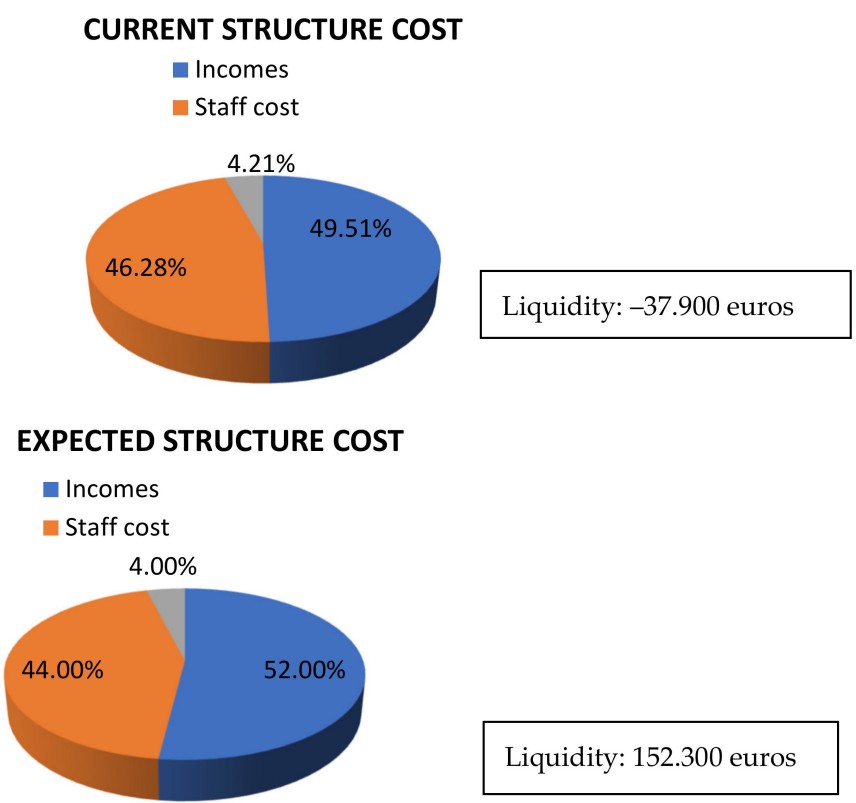

**Figure 10.** Cost structure. Source: own elaboration.

Any strategic approach must be based on a validated and proven methodology. In this specific case, we are going to work with the SMART methodology, applied to the improvement of teaching actions [74], as shown in Table 8.

**Table 8.** SMART methodology applied.

| SPECIFIC | Implementation of a digital teaching platform and virtual simulators. |
|---|---|
| MEASURABLE | Tuition of €300 in exchange for a differentiated methodology. |
| ATTAINABLE | The center has the resources to carry it out. |
| REALISTIC/RELEVANT | It is being done at other educational levels with very positive results, therefore it is feasible to implement and has a high probability of success. |
| TIME-RELATED | 1st year for implementation. |
| | 2nd year for implementation. |

SSource: Own elaboration.

This new teaching methodology based on digitalization will link the relationship of this type of educational center with the students, allowing to increase the number of students and to provide an added value that may allow to increase the fee to be paid by each student, and thus ensure the economic survival of the educational center. According to a 2017 INE report, in 97.4% of households, there is some mobile device and 3 out of 4 children aged 12 have a cell phone and 93.9% in the population aged 15 [75]. Similarly, it has been proven that these new methodologies lead to a greater perception of educational quality by students and their parents [76].

### 3.2.3. Planning and Feasibility Plan (SO3)

Every strategy requires the development of a feasibility plan to carry it out. In this case, the Vroom and Yetton matrix will be used as a planning reference [77], as shown in Figure 11.

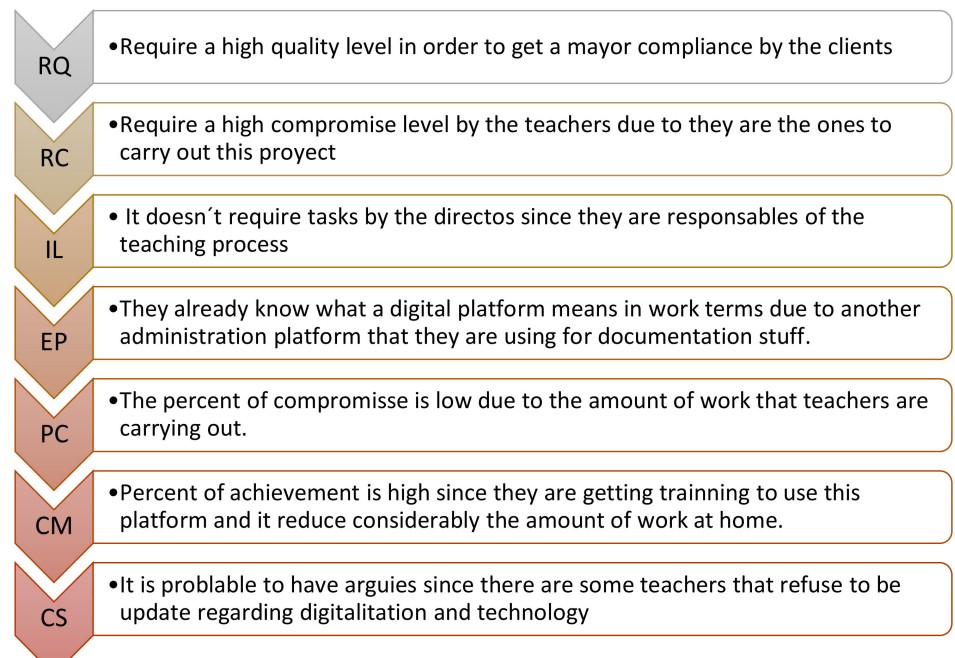

**Figure 11.** Vroom and Yetton matrix process. Source: own elaboration.

The time required would be longer than that usually needed for the incorporation of a digital platform in teaching, since it can be started sufficiently in advance and, therefore, far from being reactive, the process has time to mature and settle in. To this process should be added the training itself (it could be at the same time) in the virtual simulators. The following figure (Figure 12) shows the timing of the different phases in the incorporation of the strategic plan.

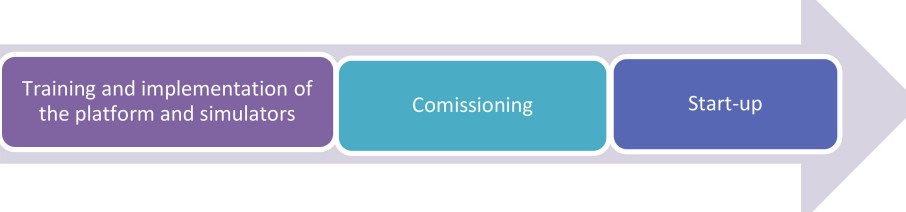

**Figure 12.** Do-check-action plan. Source: own elaboration.

The technological development involved in a digital teaching platform with virtual simulators covers various areas, from the development of servers and platforms to the development of content and activities. Likewise, it is necessary to develop the management of the platform, including the management of student data and its monitoring. Finally, the center will have to inform the students and their teachers of the existence of this new methodology, which, a priori, will mean an increase in the aforementioned fee.

## 4. Results

As has been pointed out in the development of the study, the objective of the work does not lie in the development of a business plan for an educational center. Neither does it seek to compare between an exclusively public education model or a mixed system, as in the Spanish case. This work defines a specific methodology for financial viability that allows for the sustainability of this educational center model, through a process of teaching innovation based on the application of new technologies to the educational sphere. With this objective, beyond the suitability of one model over another, the objective is to contribute academically to the line of studies linked to the spanish educational model in terms of the typology of educational centers, attempting to contribute to the line of sustainability of one of these types of centers.

Apart from the specific results that will be developed below, it is important to point out that this model of teaching innovation can be replicated in other schools. As the researcher Alfredo Hernando, author of the project "Journey to the School of the 21st century", points out, the improvement of educational processes must come from the centers and teaching teams to the system as a whole, and not the other way around, as the administrations are not capable of maintaining the development of these processes over time in a solid way. Continuing along these lines, Hernando points out that this innovation in processes must be accompanied by the sustainability, often financial, of what is proposed [78]. This contribution has been highlighted as it is directly related to the contribution made in this paper.

### 4.1. Financial Results

The results of the proposed proposal must be adapted to the financial conditions existing in these types of centers. Moodle, being a free software, is ideal since there is no need to pay for using it and it offers an excellent quality service. Proof of this is that certain public entities and renowned universities, as already mentioned, employ and train their teachers in the use of this platform [79]. As already mentioned, at this early stage of development, the virtual simulators to be used will also be free of charge.

The strictly technological investment is very small. Only a minimum investment in a hard disk with automatic backup is required (about €200). The rest of the maintenance costs would be included in the existing computer maintenance item, so there is no additional cost as described above. In anticipation of possible contingencies, a contingency fund of €1000 is established for possible unforeseen events.

With respect to the training required for the center's personnel, no major outlay will be necessary. This training will be mostly bonusable via social security contributions paid by the center. As of BOE-A-2017-7769 these types of centers are entitled to a bonus of 0.525% of the contributions, which would mean in our case study a bonus of €3695. Similarly,

the same regulation establishes that the center has to pay 10% of the cost of this training, i.e., €369 [80].

As mentioned above, it has been proposed to establish an additional 300 euro tuition fee when integrating this project. For this, the perceived value of the benefits of digitization and the application of new teaching methodologies has to reach at least a target price of €300 in the perception of students and parents. However, there is the possibility of having to reduce the fee if this increase in value is not accepted. In view of this possibility, it would be necessary to assess at what level of fee increase it would be profitable to continue with the project. This simulation is shown in Table 9.

**Table 9.** Revenue analysis.

| | Annual Income (634 Students) | Annual Fee Per User | Observations |
|---|---|---|---|
| PRICE A | €20,000.00 | €31.55 | A fee of €31.55 and 634 students would only be able to meet the annual expenses not covered by the monthly quotas and therefore would not be able to get out of financial or operational stagnation. |
| PRICE B | €70,000.00 | €110.41 | A fee of €110.41 and 634 students would be able to meet the annual expenses and get out of financial stagnation and, in the following years, would get out of operational stagnation. However, it would not be possible to reward teachers for their work on the platform, which may not be worth the effort and therefore the perceived value of the platform would be zero. Therefore, it is necessary to set aside an item of €30,000 for bonuses to teachers at the end of the school year. This could not be addressed in the first year and would jeopardize the strategy for the following year. |
| PRICE C | €100,000.00 | €157.73 | A fee of €157.73 would achieve the above objectives and a bonus for teachers, but the center would lack liquidity until the beginning of the next year's tuition collection. |
| PRICE D | €130,000.00 | €205.05 | A fee of €205 would cover the aforementioned and would also provide €30,000 of liquidity. |
| PRICE E | €190,000.00 | €299.68 | A fee of of €300 would cover the above plus €60,000 to be able to invest in the improvement of the facilities. |

Source: own elaboration.

As can be seen in Table 9, different scenarios have been considered assuming that the estimated 634 students would assume the fee increase. In this same assumption, it is shown that with a fee increase of €200 or more, the objectives set out in the case study would be achieved.

However, as has also been pointed out, the revision of the results must take into account the possibility of a decrease in the number of students as a result of the new fee. Table 10 shows an analysis of the different possible scenarios.

**Table 10.** Analysis according to the potential variation in the percentage of students.

| | Annual Income (634 Students) | Annual Fee for the Platform and Virtual Simulators per User | Observations |
|---|---|---|---|
| % Current students | €1,902,000.00 | €0 | The risk under all assumptions if the number of pupils decreases is worse than the current situation in which such a school would find itself. |
| 10% Students less | €1,882,980.00 | €300 | |
| 20% Students less | €1,673,760.00 | €300 | |
| 20% Students less | €1,464,540.00 | €300 | |
| Maximum margin of loss of students 9% | €1,902,000.00 | €300 | |

Source: own elaboration.

As can be seen from the table above, a loss of more than 9% of students, even if the remaining 91% assume an increase in the fee of €300, makes the project unfeasible. Therefore, it is of vital importance to know in advance whether the parents of the students

are willing to pay the proposed amount for the following year. For this reason, as described in the development of the methodology, it is essential to include a survey to obtain this information.

The survey data will also be used to evaluate the option of working with the platform and the virtual simulators, depending on each specific case. That is to say, if you want to work with the platform you will pay the registration fee and if not, you will not have the right to benefit from it. This means that the overall methodology in the center will not be able to be completely innovative and will have to continue with a classical methodology incorporating this new methodology as a complementary extracurricular activity. Obviously, the price must be reduced since the innovative methodology in the classroom is lost and therefore the perceived value will be considerably lower. This aspect would leave possible scenarios considering that the number of students is maintained.

The following table (Table 11) shows the income that would be obtained exclusively from the incorporation of the platform and virtual simulators if the fee were €200 depending on the percentage of students who opt for it.

**Table 11.** Analysis of income according to percentage of students.

| Percentage of Students Who Pay | Annual Revenue (634 Students) | Annual Fee per User |
|---|---|---|
| 100% | €126,800.00 | €200 |
| 75% | €95,100.00 | €200 |
| 50% | €63,400.00 | €200 |
| 25% | €31,700.00 | €200 |

Source: Own elaboration.

Therefore, if the survey is not favorable, this option is always possible, even if it does not meet the medium-term objective.

*4.2. Survey Results*

In the process of incorporating the digital platforms and virtual simulators (second quarter of the academic year (in this type of center, the period between April and June)) the students will be able to start working on these aspects at the suggestion of the teachers during the commissioning and start-up phase. During the commissioning phase, there will be specific and isolated activities to make the necessary checks and corrections. However, during the commissioning phase, which would correspond to the third quarter, the platforms and simulators must be fully operational and corrected so that as many activities as possible can be carried out with the students. The objective is to show students the benefits and advantages provided by this new teaching methodology during the school year. For this reason, it is important to collect information through the survey at the peak time, which will coincide with the month of May.

The survey is of vital importance to determine the perceived value and, therefore, to be able to establish a price for it, considering that the loss of students would be a failure of the strategy. For this purpose, two channels and two types of surveys will be established. The channels will be physical (manual survey) and online (Google Forms type form) and the recipients of these will be the students and parents/legal guardians. On the one hand, students will be given a survey evaluating their satisfaction with the center at the educational level and then a survey evaluating the same concepts, but with respect to the center with the use of digital platforms. In order to measure the results, the questions should be adapted as far as possible to satisfaction surveys conducted by public agencies for all educational centers. In the case study, since a model of an educational center located in the municipality of Granada has been chosen, the public bodies of reference will be the Junta de Andalucía and the Granada city council itself. In those cases where there are no valid references, self-made references will have to be used. The age ranges for carrying out the survey are from 1st ESO (Compulsory Secondary Education) to 2nd year of Bachillerato. For the primary and infant education stages, only three questions would be asked (one

per indicator) and they would be the following: do you like the class more with digital platforms? Does the use of digital platforms make you want to go to class more? Do digital platforms help you to learn more? With a "YES/NO" type of answer. That is why the assessment will be done by cycles and will be classified in the same way.

4.2.1. Student Satisfaction Survey

In an initial survey, reference values of satisfaction with respect to the center will be obtained in various aspects of interest related to the indicators that will be used to obtain information of interest in the decision-making process of the strategy in relation to the price and form of enrollment for the use of digital platforms (see Appendix A, Table A1. Input survey for analyzing the indicators).

Next, we present the survey to be conducted by the students once they have experienced teaching with digital platforms during the commissioning period (see Appendix A, Table A2. Exit survey to analyze the indicators).

From the comparison of the previous surveys, relevant information can be obtained regarding the relationship between price and the value perceived by users. This comparison would be carried out by calculating the weighted average of the score obtained for each question in each survey, thus quantifying in points the difference between one type of teaching and another in the different indicators of interest:

- Student satisfaction
- Improvement of school failure rates
- Quality of the material used in the digital platforms.

When contrasting the data, several scenarios can be identified, among which we highlight the following (Table 12):

- 1st case: all three indicators reflect a significant improvement.
- 2nd case: some indicators reflect a significant improvement.
- 3rd case: no indicator shows a significant improvement.

The purpose of the surveys aimed at students is both the evaluation of the system by end users and the possibility that this positive perception will be perceived by parents and increase their willingness to pay the required fee. Similarly, questions focused on measuring school failure have been included, since it represents a considerably high percentage (21.9% as already indicated) of students and their involvement will be fundamental for the successful development of the project (as was verified, if a percentage higher than 9% were to drop out of the center due to the fee increase, the strategy would not be consistent).

4.2.2. Parent Satisfaction Surveys

The first part of the survey is related to the value perceived by parents through their children's perceptions of the use of virtual platforms and simulators (see Appendix A, Table A3). Survey for parents/legal guardians). This first part is fundamental since it indicates the feedback received by the parents from the students. It also allows us to know the degree of involvement of the parents in the educational process of the center.

The second part consists of analyzing the perceived value in economic quantification, that is, how much they would value their children having the possibility of working digitally and with innovative methodologies.

The third part of the survey responds to the possibility or not of parents paying the fee. As already mentioned, a maximum of 10% uncertainty can be assumed, i.e., if more than 9% of respondents have not received information from either the center or the students, it is not possible to impose tuition because the consequences could be worse than the initial situation. In this case, optional enrollment for the use of the platform and simulators would be chosen. If, on the other hand, a scenario of parental participation and information appears, the results of the survey will be considered. The sections of the interview with a control element will be questions three and four. Question three assesses the perceived

value numerically and question four, monetarily. Therefore, surveys with non-relational values will not be considered.

**Table 12.** Analysis of possible scenarios.

| Positive Indicators | Strategy |
|---|---|
| Student Satisfaction | This is the most significant aspect of the survey. However, it indicates that students in a situation of school failure have not been motivated and that the material of the platforms and the application of the simulators should be improved since they do not see any added value. In this case, their use would be considered optional. |
| Improvement of school failure rates | This index would only help us in the case of school dropouts (21.9% in Andalusia as a reference for our case study). In the case of on-site students, the objectives indicated would not have been met, but it would serve to publicize it with a view to attracting students who wish to take online training. The measure to be followed is to promote the methodological innovation as an option in the enrollment under the assumption that the main consumers of it will be those students at risk of school failure. |
| Quality of the material used in the platforms and simulators | This is the indicator that reflects the success of the correct execution of the strategy and is therefore valid; however, it shows that the dissatisfaction of the students comes from other aspects to be analyzed. As it has not led to an overall improvement, it is not convenient to incorporate the mandatory payment in the enrollment, but to raise it as optional, since the surveys predict a perceived value by the students of the use of the same. |
| Student satisfaction: improvement of school failure rates | This situation indicates that the quality and the system need to be improved, but that the methodology used is positive. This is perhaps the most doubtful combination among the positive ones, since it has improved the situation but has failed in the key part of this improvement. Therefore, as it is an incongruent situation, it is not possible to make an assessment. For this reason, enrollment would be optional and for the following year we will work on improving the digital material until the expected ranges of satisfaction are obtained. |
| Special Case 2 Student satisfaction: quality of the material used on the platforms and simulators | These results would be, a priori, the most logical. Normally, school failure cases are indifferent to one didactic system or another. This is why it can be considered as satisfactory and therefore contains the added value. In this situation, the payment of the tuition fee is considered mandatory, but as it has not reached total satisfaction, the price will be €200 and the same payment options as those described above will be proposed. |
| Improvement of school failure rates: quality of the material used in the platforms and simulators | This situation indicates that the platforms and simulators help the teaching process and motivate students to continue working. However, since there is no improvement in satisfaction, it indicates that it is the application part that is not working. Therefore, the measure to be carried out would be the option to pay for the use of the digital platform and the improvement in the following course of the methodology in the use of the same until the objective is achieved. |
| Special Case 3 Student satisfaction: quality of the material used in the digital platforms and simulators; improvement of school failure rates. | This situation indicates that digital platforms help the teaching–learning process and motivate students to continue working. It also means an improvement in satisfaction. Therefore, the measure to be carried out would be the obligatory payment for the use of the platform and virtual simulators, having achieved the proposed objective. |

Source: Own elaboración.

The indexes represented in each section of the questionnaire will be considered as follows: perceived value, monetary valuation, economic commitment of the client.

A) Customers have a high perceived value, a high monetary valuation and a high payment commitment. This is the most favorable situation, and therefore, the license plate would be imposed.

B) Customers have high perceived value, high monetary valuation and low financial commitment. In this case, it indicates that they are happy with the proposal but do not want to pay more to the center for their children's education. That is why the option to take is to offer optional tuition on digital platforms.

C) Customers have a high perceived value, a low monetary valuation and a high/low economic commitment (not contemplated due to incongruence for the case).

D)   Customers have low perceived value, high monetary valuation and high/low economic commitment (not contemplated due to inconsistency for the case).

Options C and D will be those in which the control elements do not give consistent results. For the rest of the cases, given the risk involved, the optional matrix will be proposed.

*4.3. Global Results*

As has already been pointed out above, beyond the results that can be applied to a specific school and subsequently replicated in other schools, this study aims to provide global conclusions that allow us to expand the existing literature in the field of the Spanish educational model. Beyond assessing the preference of a model based on a greater or lesser presence of private schools, it is necessary to evaluate their contribution to the overall system. Various studies (in our case, the one carried out by the School Council of the Community of Madrid in 2019) place the average educational expenditure of a pupil in public education at approximately €4505 per pupil and at €3908 per pupil in the case of subsidized education (private education financed with public funds) [81].

On the other hand, the costs to the public purse of pupils in private schools are almost insignificant, as the fees come almost entirely from parents' payments. Only through possible tax deductions depending on each autonomous community (in the case of Spain) is there a reduction in the public funds associated with private education (these deductions do not, in any case, exceed the amount of €800 per pupil in private education). Thus, assuming that the tax deduction is applicable throughout Spain, and that all families who choose a private school could benefit from it, the estimated savings in public expenditure on education per private school pupil would be around €3000. Only in the case of the center studied in the study would the 700 pupils referred to above represent an estimated saving of €2,100,000. If we apply this same equation to all students in private pre-university schools, the figure would be much higher. According to the latest data provided by the Ministry of Education (2020), the total number of students enrolled is 8,083,994, of which 31.7% are in private schools. In other words, there would be around 2,662,626 pupils in private schools. If we apply the estimated average savings expenditure to this total number, we find a decrease in public spending on education of around €7,687,878,291 due to the existence of public schools. In other words, public expenditure on education would increase by more than seven thousand six hundred million € if the educational model described in this paper did not exist. In order to put this figure into perspective, it is interesting to evaluate the overall data on educational expenditure, as can be seen in Figure 13.

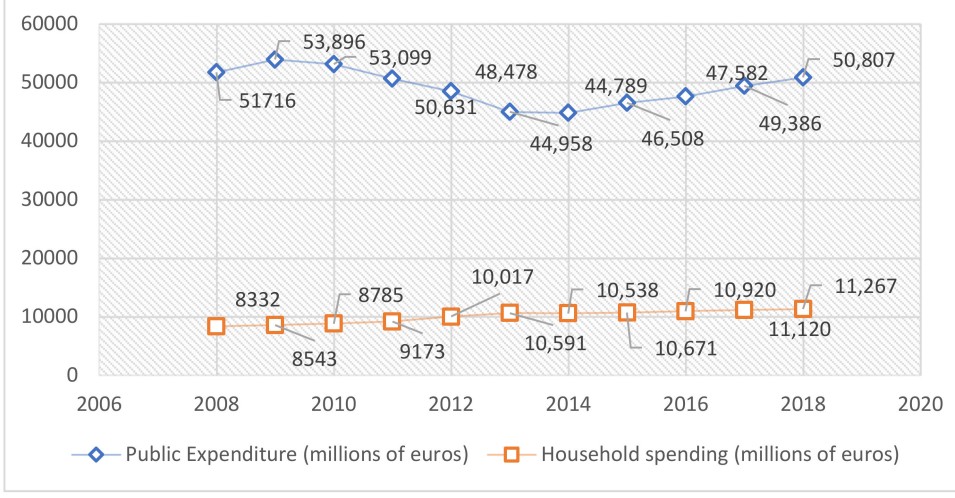

**Figure 13.** Public expenditure and household spending in education. Source: own elaboration based on Ministry of Education.

As can be seen, in the total public expenditure on education, the savings calculated above represent a percentage of more than 15% of the overall educational expenditure by the administrations. On the other hand, according to the latest updated data from the Ministry of Education for the academic year 2018/19, there are 205,241 teachers outside the public sector in Spain. Apart from the estimated direct impact in terms of expenditure per pupil, it would be necessary to add the impact generated by this number of teachers who would change their current situation from being remunerated by a private entity to being directly employed by the public system.

In summary, therefore, it seems clear that the non-existence of this type of educational center (for various reasons, including the financial sustainability discussed in this paper) would mean that the entire current educational system would need to be remodelled on the basis of its sustainability.

## 5. Discussion

As already indicated in the paper, as in other countries, Spain has different models of educational institutions. This distinction can be made both in terms of the type of owner (public and/or private) and in terms of the source of funding (public and/or private). In all the above combinations, the model can give rise to three types of schools: public, private and publicly funded private. Logically, the appropriateness or otherwise of these types of models has been widely debated by the scientific community.

With all that has been said above, in order to definitively link the introduction of the article with the future objectives, results and the research methodology itself, it is important to point out that the main objective is, as has already been indicated, to analyze the financial viability of a very representative educational center model in Spain: a private, non-subsidized center. This viability is not carried out with the sole intention of validating a business model but with the intention of assessing its importance in the global educational context, as a guarantor of its sustainability and its contribution to quality educational standards. In order to do so, it is interesting to start from previous academic studies that have dealt with these two aspects at a global level:

- Importance of this type of educational institution in the global context
- Contribution to the financial viability of the system and to the quality of education.

As already pointed out in Section 2, once the historical context of the Spanish educational model has been defined, it is necessary to carry out a literature review of the positions in favor and against this model. As authors such as Touriñán report, the debate between public and private education has always been a debate in a country like Spain, becoming a debate not only in the educational sphere but also in the political sphere [82]. In fact, with percentages that tend to oscillate between 70% and 30%, respectively of public and private education, out of the total with a temporary oscillation range of around 10%, as authors such as Ortega or Mínguez indicate, the private education model is a reality that coexists with the public system, complementing it and offering different alternatives to parents [83].

Logically, this debate has had its detractors and its defenders. Basically, the academic debate has been divided into two opposing positions:

- Usually, the arguments put forward by different authors regarding the inappropriateness of a private education system are due to the supposed discrimination it produces on those families who cannot afford this educational model in economic terms (the contributions of authors such as Villarroya Planas and Escaedibul Ferrá (2008), Bernal and Lorenzo (2013), Ball and Youdell (2007), etc., *although included in the references, are also included in this section for a more structured reading*). Here, various authors as Villarroya Planas and Escaedibul Ferrá have related the importance of family factors such as the educational and cultural level of the parents, or their own socio-economic situation or nationality when choosing the educational center model [84]. In the same vein, different authors as Bernal and Lorenzo directly define private education in Spain as a source of inequality [85], and even, according to authors such as Ball and

Youdell (2007), as a change in the model that sees education as a private good that serves the interests of the economy [86]. Finally, within this field of studies, many authors (Ball; Van Zaten; Bernal; Olmedo; Alegre) have reflected on and questioned the supposed advantage of the mixed educational model when it comes to favoring competitiveness and the parents' capacity to choose an educational center [87–91]. These authors show that the supposed capacity of choice is almost always reduced to the middle and upper classes, as the less economically able classes choose to take their children to the school closest to their place of residence.

- On the other hand, authors who have advocated this mixed model of education system as optimal have based their arguments on the good academic results of private schools and the positive effects on the overall education system by allowing many students not to be taken over by public education, with the consequent increase in the cost of education (as noted above, the contributions of these authors (Feito, Betts, Evans, Schwab, etc.) are also included in the references, although they are also introduced in this section). In this sense, several authors such as Feito (2002) have defended the need to incorporate higher levels of competitiveness into the educational sphere by proposing new scenarios for action, including market scenarios [92]. Traditionally, many authors (Betts; Evans and Schwab; Sander) have argued that high quality standards in private education have an important positive effect on future labor relations [93–95]. More contemporary studies focused on the Spanish reality (Bonal; Calero) have presented the mixed education system as a guarantee for the economic sustainability of the public education system [96,97]. These authors argue that the private model allows the public system to avoid taking on an even larger number of students, which would make its financial sustainability very difficult.

Therefore, it seems clear that the main reluctance to the school model studied in this study (private social schools) is due to the cultural and socio-economic differences of the students themselves. In this sense, authors such as Bloosfeld and Shavit point out that the expansion of the educational system favors an educational selection based on meritocracy, which can reduce or increase previous inequalities depending on the effort of the students [98]. In the same vein, authors such as Arum point out that in countries with a large private school sector, public schools perform better than average [99]. However, authors such as Coleman, Hoffer and Kilgore argue that, systematically, private schools tend to provide a better education for their students than public schools and that these public schools are usually chosen by students from low socio-economic families [100].

In conclusion, it can be said that the academic discussion of the subject in question is wide-ranging and heterogeneous. As authors such as Fernández Anguita point out, "private education in Spain is not so private as it has to comply with the conditions of regulated education" [101]. However, the aim of this paper is not to assess the suitability of one model or another, but to define a methodology applicable to private social education centers that allows the current model to be maintained from a financial point of view, with its advantages and possible limitations. It should be noted that this situation is not exclusive to the Spanish case. The Program for the Promotion of Educational Reform in Latin America and the Caribbean itself states as one of its objectives "the identification of policies that help to ensure that private education meets the educational objectives of equity, access, quality, research and benefit for clients" [102].

Beyond pointing out these academic approaches at a global level, the present work is based precisely on analysing the viability and sustainability of a specific educational model (private social education). In this way, it will contribute to taking measures for its financial viability, which will allow the advantages of this type of center in the spanish educational model to be maintained.

## 6. Conclusions

The conclusions reached in this study go beyond the business model from a business point of view. The importance of the financial part developed lies in the fact that it will be

a point of support for the sustainability of the educational center model analyzed. In other words, the main aim of the work is not to present a business model, but to enable this type of educational center to continue providing its services within the Spanish educational model. The aim of this work is to contribute to the existing literature on the current educational system in Spain by analyzing the viability of one of the most representative centers. In other words, the importance lies in the fact that without the existence of private social centers, the current system would have to change, without analyzing whether this change would be positive or negative on a global level.

As shown in Section 4, the existence of private schools represents a cost saving of around 15% of overall educational expenditure. In most cases, because of market criteria, the non-viability of private school models depends on their financial situation. This work helps, firstly, the financial viability of this type of center and, secondly, the overall viability of the system.

Similarly, it is important to take into account the figure of 205,241 teachers outside the public sector in Spain. By means of innovative teaching methodologies, a methodological proposal is put forward to ensure that this volume of teachers continues to work within the current model.

Logically, the limitations of this research are important and go beyond the results obtained. Thus, the current educational model in Spain ultimately depends on the decisions taken by the public authorities. In other words, political ideologies have a strong influence on the decision to favor one educational model or another. Logically, this paper has tried to reflect a current reality from an academic point of view, regardless of whether this context can be changed by the administration. In the same way, it presents the limitations inherent to academic work in which the methodology of the case analysis is used. Although this aspect has already been referred to in Section 3.2, logically, the application of a specific case study to different realities must be carried out with caution and taking into account the particular situations that appear in each specific case.

Turning to the specific case of this paper, as a general rule for this type of educational centers, especially in the existing academic distribution in a country like Spain, financial risks jeopardize their ability to perform their work. With this work it has been possible to provide a financial solution accompanied by a new teaching methodology that includes a process of digital transformation and educational innovation through the incorporation of digital platforms and virtual simulators. With the work developed, a specific model for a standard center is adapted through a case study methodology, but it can easily be adapted in a generic way to the peculiarities of any educational center of this model, guaranteeing its survival both from a financial point of view and from an operational point of view. As seen in the previous section, in the event that the information received in the surveys is satisfactory, the strategy to be followed is clearly detailed in each of the possible scenarios. The situations that would be generated according to the different possible scenarios would be as follows:

1.  Scenario A: The center obtains 25% of clients with a quota of €200 of digital platform, which means an income of €31,700.00. In this case, the extra annual expenditure would be met and therefore a margin of maneuver of around €15,000 would be obtained.
2.  Scenario B: The center obtains 50% of its clients with a €200 digital platform fee, which means an income of €63,400.00. In the first place, the extra annual expenses would be met, an important part of the center's debt would be paid off, and there would be a margin of €23,400.00 for teacher bonuses and working capital.
3.  Scenario C: The center has 75% of its clients with a €200 digital platform fee, which means an income of €95,100.00. First of all, the extra annual expenses would be met and 100% of the debt would be paid. This would completely address the operational and financial stagnation of the center. With the remaining €30,000 we would be in a position to reward the teachers and have a comfortable working capital.
4.  Scenario D: The center obtains 100% of customers with a €200 digital platform fee, which means a profit of €126,800.00. With these profits, all the economic objectives of

the strategy would be achieved, reducing 100% of the debt. Teachers would receive a bonus with the proposed maximum of €30,000, and a working capital of €21,800 would be available.

5. Scenario E: The center obtains 100% of clients with a quota of €250 of digital platform, which means a profit of €158,500.00. With this situation, all the objectives already marked in situation D would be fulfilled, increasing the working capital to €53,500.

6. Scenario F: The center obtains 100% of clients with a €300 digital platform quota, which means a profit of €189,600.00. With this situation, all the objectives already set in situation D would be met again, increasing the working capital to €84,600.

With this excess income, depending on each of the scenarios, it will be possible to meet the total extra expenses, assume the operating expenses and, depending on the situation, supplement the teaching staff in the region of €30,000 per year. In the same way, as the forecasts improve, investments can be made to improve the center as a whole. In the same way, the work includes the necessary methodology so as not to approach this methodological process until the appropriate conditions are met.

In short, this study has sought to propose a methodology to analyze the feasibility of different strategies to respond to the problem that occurred in the analysis of the case, i.e., the problems of financial sustainability of educational centers.

It is also worth highlighting the importance of the teacher in the whole process described, as it has been the global improvement in all aspects that has allowed the improvement of the situation of the center and the educational perception on the part of students and parents. This methodological improvement in these types of centers should be the first step to be able to adapt teaching to a growing digital environment that results in a teaching innovation, an improvement in the educational level of the center, an improvement in the satisfaction perceived by students and parents and a suitable tool to reduce school failure rates. This improvement proposal would involve implementing flipped classroom techniques, gamification, project work, etc., based on the use of virtual simulators. This is a proposal for continuous improvement that would involve carrying out training for teachers during the commissioning phase in order to achieve all the possibilities offered by this digitization process, ensuring the financial viability of the center.

As a final conclusion, it should be pointed out that the final objective of the work does not have to do with drawing up an effective business plan for a specific educational center. It is about proposing a methodology that allows for the financial sustainability of a very common educational center model in Spain, based on technological innovation. With this sustainability, we are proposing, from an academic point of view, the possibility of maintaining the mixed education system that currently exists. In the same way, as has already been pointed out, once these processes of teaching innovation have been developed and consolidated, they will serve to adapt their use to the education system as a whole, obtaining an overall improvement in the system. In the same way, this work has also highlighted the academic contributions regarding the impact of the application of new technologies in the field of education, explaining their advantages and the possibility of their application.

Possible extensions to this work should be linked to a greater detailing of the impact of each educational model on the educational system in general, with emphasis on the contributions made by these types of schools. Similarly, it would be advisable to study the results indexes by the type of school according to the socio-economic characteristics of the pupils enrolled.

**Author Contributions:** Formal analysis, R.F.R.F., J.J.-V., S.L.N.A. and R.S.V.; conceptualizacion, R.F.R.F. and J.J.-V., investigation, R.F.R.F. and J.J.-V.; methodology, R.F.R.F. and R.S.V., supervision, R.F.R.F. and S.L.N.A.; writing—review & editing, R.F.R.F. and J.J.-V. All authors have read and agreed to the published version of the manuscript.

**Funding:** The APC was partially funded by the incentive granted to the authors by the Catholic University of Ávila and by the incentive granted to Prof. Phd. Sergio Luis Náñez Alonso by MDPI.

**Institutional Review Board Statement:** Not applicable.

**Informed Consent Statement:** Not applicable.

**Data Availability Statement:** Not applicable.

**Conflicts of Interest:** The authors declare no conflict of interest.

## Appendix A

**Table A1.** Input survey to analyze the indicators.

| | Initial Evaluation | | | | | | | | | | | |
|---|---|---|---|---|---|---|---|---|---|---|---|---|
| | **Regarding Student Satisfaction** | | | | | | | | | | | |
| 1 | How satisfied are you with the explanations of the topics you receive? | 1 | 2 | 3 | 4 | 5 | 6 | 7 | 8 | 9 | 10 | Andalusia |
| 2 | How motivating are the activities proposed by the teachers? | 1 | 2 | 3 | 4 | 5 | 6 | 7 | 8 | 9 | 10 | Andalusia |
| 3 | Globally, how do you value the way the teachers give the classes? | 1 | 2 | 3 | 4 | 5 | 6 | 7 | 8 | 9 | 10 | Andalusia |
| 4 | How do you value the information received on how to evaluate your learning in the different subjects? | 1 | 2 | 3 | 4 | 5 | 6 | 7 | 8 | 9 | 10 | Andalusia |
| 5 | How do you value the teachers' way of evaluating? | 1 | 2 | 3 | 4 | 5 | 6 | 7 | 8 | 9 | 10 | Own |
| 6 | Do you feel listened to and cared for by the teachers? | 1 | 2 | 3 | 4 | 5 | 6 | 7 | 8 | 9 | 10 | Own |
| 7 | Do you enjoy attending class? | 1 | 2 | 3 | 4 | 5 | 6 | 7 | 8 | 9 | 10 | Own |
| 8 | I am satisfied with the education I have received so far. | 1 | 2 | 3 | 4 | 5 | 6 | 7 | 8 | 9 | 10 | Granada |
| 9 | The course meets the initial expectations I had at the beginning of the course. | 1 | 2 | 3 | 4 | 5 | 6 | 7 | 8 | 9 | 10 | Granada |
| 10 | The contents of the courses are adequate | 1 | 2 | 3 | 4 | 5 | 6 | 7 | 8 | 9 | 10 | Granada |
| 11 | The classes help me to improve my study skills. | 1 | 2 | 3 | 4 | 5 | 6 | 7 | 8 | 9 | 10 | Granada |
| 12 | I can identify the contents given in class in my daily life. | 1 | 2 | 3 | 4 | 5 | 6 | 7 | 8 | 9 | 10 | Granada |
| 13 | I feel comfortable in the classroom during the classes. | 1 | 2 | 3 | 4 | 5 | 6 | 7 | 8 | 9 | 10 | Granada |
| | **Concerning School Failure** | | | | | | | | | | | |
| 14 | The teachers have a positive influence on my desire to study. | 1 | 2 | 3 | 4 | 5 | 6 | 7 | 8 | 9 | 10 | Own |
| 15 | The methodology at the center positively influences my desire to want to study. | 1 | 2 | 3 | 4 | 5 | 6 | 7 | 8 | 9 | 10 | Own |
| 16 | Would you drop out of school if you could? | 1 | 2 | 3 | 4 | 5 | 6 | 7 | 8 | 9 | 10 | Own |
| | **Regarding the Resources Used** | | | | | | | | | | | |
| 17 | The didactic material used facilitates the learning of the subject. | 1 | 2 | 3 | 4 | 5 | 6 | 7 | 8 | 9 | 10 | Granada |
| 18 | The books used clarify possible doubts | 1 | 2 | 3 | 4 | 5 | 6 | 7 | 8 | 9 | 10 | Granada |
| 19 | I find the classes motivating and entertaining. | 1 | 2 | 3 | 4 | 5 | 6 | 7 | 8 | 9 | 10 | Granada |
| 20 | My level of attention during the classes is | 1 | 2 | 3 | 4 | 5 | 6 | 7 | 8 | 9 | 10 | Own |
| 21 | When I do my homework, I easily remember what I have worked on in class. | 1 | 2 | 3 | 4 | 5 | 6 | 7 | 8 | 9 | 10 | Own |

Source: Own elaboration.

**Table A2.** Exit survey to analyze indicators.

| | Final Evaluation | | | | | | | | | | | |
|---|---|---|---|---|---|---|---|---|---|---|---|---|
| | **Referentes a La Satisfacción Del Alumno** | | | | | | | | | | | |
| 1 | How satisfied are you with the explanations of the topics you have received in class with the use of digital platforms and simulators? | 1 | 2 | 3 | 4 | 5 | 6 | 7 | 8 | 9 | 10 | Andalusia |
| 2 | How motivating are the activities proposed with the use of digital platforms and simulators? | 1 | 2 | 3 | 4 | 5 | 6 | 7 | 8 | 9 | 10 | Andalusía |
| 3 | Globally, how do you value the way the classes are given with the digital platforms and simulators? | 1 | 2 | 3 | 4 | 5 | 6 | 7 | 8 | 9 | 10 | Andalusía |
| 4 | How do you value the information received on how to evaluate your learning with the use of digital platforms and simulators? | 1 | 2 | 3 | 4 | 5 | 6 | 7 | 8 | 9 | 10 | Andalusía |
| 5 | How do you value the way of evaluating with the use of digital platforms and simulators? | 1 | 2 | 3 | 4 | 5 | 6 | 7 | 8 | 9 | 10 | Own |
| 6 | Do you feel listened to and attended with the use of digital platforms and simulators? | 1 | 2 | 3 | 4 | 5 | 6 | 7 | 8 | 9 | 10 | Own |
| 7 | Do you enjoy attending class with the use of digital platforms and simulators? | 1 | 2 | 3 | 4 | 5 | 6 | 7 | 8 | 9 | 10 | Own |
| 8 | I am satisfied with the education provided by the use of digital platforms and simulators. | 1 | 2 | 3 | 4 | 5 | 6 | 7 | 8 | 9 | 10 | Granada |
| 9 | The use of digital platforms and simulators meets the initial expectations I had at the beginning of the course. | 1 | 2 | 3 | 4 | 5 | 6 | 7 | 8 | 9 | 10 | Granada |
| 10 | The contents of the digital platform and simulators are adequate. | 1 | 2 | 3 | 4 | 5 | 6 | 7 | 8 | 9 | 10 | Granada |
| 11 | The classes with the digital platform and simulators help me to improve my study work. | 1 | 2 | 3 | 4 | 5 | 6 | 7 | 8 | 9 | 10 | Granada |
| 12 | I can identify the contents given with the digital platform in my daily life. | 1 | 2 | 3 | 4 | 5 | 6 | 7 | 8 | 9 | 10 | Granada |
| 13 | I feel comfortable in the classroom during the classes with the use of digital platforms and simulators. | 1 | 2 | 3 | 4 | 5 | 6 | 7 | 8 | 9 | 10 | Granada |

**Table A2.** *Cont.*

| | Final Evaluation | | | | | | | | | | | |
|---|---|---|---|---|---|---|---|---|---|---|---|---|
| | Regarding School Failure | | | | | | | | | | | |
| 14 | Teachers with the use of digital platforms and simulators positively influence my desire to want to study. | 1 | 2 | 3 | 4 | 5 | 6 | 7 | 8 | 9 | 10 | Own |
| 15 | The methodology employed with the use of digital platforms and simulators positively influences my desire to want to study. | 1 | 2 | 3 | 4 | 5 | 6 | 7 | 8 | 9 | 10 | Own |
| 16 | Would you stop studying if you could even using digital platforms and simulators as a method? | 1 | 2 | 3 | 4 | 5 | 6 | 7 | 8 | 9 | 10 | Own |
| | Regarding the Resources Used | | | | | | | | | | | |
| 17 | The use of digital platforms and simulators makes it easier for you to learn the subject matter. | 1 | 2 | 3 | 4 | 5 | 6 | 7 | 8 | 9 | 10 | Granada |
| 18 | The digital platform and simulators used clarify possible doubts. | 1 | 2 | 3 | 4 | 5 | 6 | 7 | 8 | 9 | 10 | Granada |
| 19 | I find the classes with the use of digital platforms and simulators motivating and entertaining. | 1 | 2 | 3 | 4 | 5 | 6 | 7 | 8 | 9 | 10 | Andalusia |
| 20 | My level of attention during the classes where digital platforms and simulators are used. | 1 | 2 | 3 | 4 | 5 | 6 | 7 | 8 | 9 | 10 | Own |
| 21 | When doing homework, I easily remember what I have worked on in classes where digital platforms and simulators are used. | 1 | 2 | 3 | 4 | 5 | 6 | 7 | 8 | 9 | 10 | Own |

Source: Own elaboration.

**Table A3.** Survey of parents/legal tutors.

| | Survey of Parents/Legal Tutors | | |
|---|---|---|---|
| | Regarding Perceived Value | | |
| 1 | Has your child received any information regarding the use of digital platforms and the new methodologies of the center? YES     NO     N/A | | Own |
| 2 | Have you received any information from the center regarding the use of digital platforms and the new methodologies of the center? YES     NO     N/A | | |
| 3 | How do you value the use of digital platforms and innovative methodologies in the center? 1  2  3  4  5  6  7  8  9  10 | | Own |
| 4 | What monthly price would you put on the use of this type of platform per student? €0          €10          €10–€20          €20–€30          €30 < | | Own |
| | Regarding Their Economic Capacity | | |
| 5 | Would I be able to take on an additional monthly fee in exchange for this service? | | Own |
| 6 | If the price were higher than ________ I would consider the option of taking my child to another center. a) 10 €          b) 20 €          c) 30 € | | |
| 7 | It seems reasonable to me to contribute monthly ____ in exchange for this methodology to be implemented. a) 10 €          b) 20 €          c) 30 € | | Own |

Source: Own elaboration.

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
