# Peer review of "Methodology to Evaluate Economic Viability Plans and Digitalization Strategies in Private Social Education Centers"

_education, doi:10.3390/educsci11040170_

Round 1

Reviewer 1 Report

1.The title of the article must be shortened and clear.

2.The presentation format of the picture is wrong and needs to be corrected.

3.It is recommended that authors improve the presentation of English to make it easier for readers to understand the article.

4.There is a problem with the format of some cited references, it is recommended to improve.

5.The connection between research methods and results must be emphasized.

6.The research conclusion is clear, but is there any inference or hidden meaning. Please author must think clearly and add an explanation in the article.

Author Response

Esteemed reviewer, thank you very much for your efforts in reviewing the work and for your recommendations. They have undoubtedly contributed to our awareness of the need for improvement. In an attempt to respond to your comments, the following aspects have been corrected:

  • The title of the article has been changed to make it clearer and shorter.
  • The format of the images has been changed to adapt it to the requirements of the publication.
  • The english has been revised to try to make the article more understandable for the reader.
  • Significant contributions have been included in the introduction, research methodology, results and conclusions to try to homogenise these aspects, especially those referring to the connection between research methods and results.
  • An attempt has been made to respond in the conclusions to the limitations of the work itself and to clearly express the contributions achieved.

All these changes are highlighted in yellow to make them easier to find.

Once again, thank you very much and we hope that we have been able to reflect your contributions in the text.

Reviewer 2 Report

The paper analyzes a very important subject, the digital and financial transformation of educational systems. The paper in its present form more like a business plan or a consultative work. The introduction is merged with the literature review. No propositions/hypotheses are formed, the reflection on previous studies is missing. The introduction rather should give reader an overview of the research problem, the gap in the literature, the method and briefly the contributions. 

The rest of the paper is also not presented in usual, academic journal form, which is odd. 

The following is suggested to authors to improve their paper:

  • The authors should re-review the relevant literature and summarize the studies (for example, in a table)
  • make your contributions explicit. In doing so, authors should align  objectives with the conclusions as they don’t seem to be linked well at the moment
  • clarify the managerial, policy, and theoretical implications of this research
  • discuss limitations of the research

Author Response

Esteemed reviewer, thank you very much for your efforts in reviewing the work and for your recommendations. They have undoubtedly contributed to our awareness of the need for improvement. In an attempt to respond to your comments, the following aspects have been corrected:

  • A section called "Global Discussion" has been introduced in the introduction, in which an attempt has been made to analyse previous studies with respect to the proposed work. This section reflects on the different academic perspectives on the subject of the paper. This entire section has been introduced in a specific sub-section to facilitate its reading and to assess the value of this contribution.
  • Significant contributions have been included in the introduction (as mentioned above), in the methodology, in the results and in the conclusions in order to try to align the objectives with the final conclusions. In this way, an attempt has been made to link the methodology of the work to the final objective of the research (to assess the financial viability of this type of educational centre in order to guarantee the sustainability of the current Spanish educational system), differentiating it from the creation of a simple business plan or business model.
  • An attempt has been made in the conclusions to respond to the limitations of the work by relating it to the political and methodological implications that may limit its implementation in other centres similar to the one proposed.

On the other hand, in an attempt to improve the overall consideration of the work:

  • The title of the article has been changed to make it clearer and shorter.
  • The format of the images has been changed to adapt it to the requirements of the publication.
  • The english has been revised to try to make the article more understandable for the reader.

All these changes are highlighted in yellow to make them easier to find.

Once again, thank you very much and we hope that we have been able to reflect your contributions in the text.

Reviewer 3 Report

The article significantly lacks a discussion section. A relatively successful theoretical part describing the general conditions of education and development of human capital in the Spanish education system in the context of global - and especially European - education system with implications for social spheres, educational opportunities and funding was not logically linked to empirical research, both parts are mechanically added to each other without a deeper contextual connection, here I see a reserve for completion, deepening an otherwise interesting article.

Author Response

Esteemed reviewer, thank you very much for your efforts in reviewing the work and for your recommendations. They have undoubtedly contributed to our awareness of the need for improvement. In an attempt to respond to your comments, the following aspects have been corrected:

  • A section called "Global Discussion" has been introduced in the introduction, in which an attempt has been made to analyse previous studies with respect to the proposed work. This section reflects on the different academic perspectives on the subject of the work. This discussion section has taken into consideration aspects that relate the current spanish educational model to key variables such as the economic level of families, social spheres and equal opportunities. This entire section has been introduced in a specific sub-section to make it easier to read and assess this contribution.
  • Significant contributions have also been included in the methodology, in the results and in the conclusions in order to try to align the objectives set with the final conclusions. In this way, an attempt has been made to link the methodology of the work to the final objective of the research (to assess the financial viability of this type of educational centre in order to guarantee the sustainability of the current Spanish educational system), differentiating it from the creation of a simple business plan or business model.
  • An attempt has been made in the conclusions to respond to the limitations of the work by relating it to the political and methodological implications that may limit its implementation in other centres similar to the one proposed.

On the other hand, in an attempt to improve the overall consideration of the work:

  • The title of the article has been changed to make it clearer and shorter.
  • The format of the images has been changed to adapt it to the requirements of the publication.
  • The english has been revised to try to make the article more understandable for the reader.

All these changes are highlighted in yellow to make them easier to find.

Once again, thank you very much and we hope that we have been able to reflect your contributions in the text.

Round 2

Reviewer 1 Report

You seem to have corrected all the comments in the first round.
However, regarding English, if the paper authors does not have native English speakers, it is recommended that the manuscript be edited by a professional English editing in order to maintain the standard of the journal.

Generally, the author will be required to provide proof of editing the paper in native English.
I can’t judge this point, but leave it to academic editor to make a decision.

good luck!

Author Response

Thank you very much for your contributions to improve the work.

  • In this latest edition, we have tried to improve the work as a whole (all changes are highlighted in blue to make them more visible).
  • Similarly, we have tried to improve the English editing.

Again, thank you very much for the effort.

Best regards

Reviewer 2 Report

Thank you for your revised manuscript. I still find it ambiguous, let me provide some examples. 

1) I believe the Literature review should be an independent section. Nevertheless, "Global discussion on the co-education model in Spain" is inappropriate for a chapter title. Global discussion can be about co-educational models in general, however, I disagree that Spain would be in the focus of those discussions. You may start describing the - indeed - global findings, then, you may narrow it down to Spain. Either way, this needs some elaboration.

2) the literature review is superficial. You say, for example, "Usually, the arguments put forward by different authors regarding the inappropriateness of a private education system are due to the supposed discrimination it produces on those families who cannot afford this educational model in economic terms." Who are those authors? When did they say it? Scientific papers must have validity, hence it is critical to rely on former knowledge/studies.

I could continue the list, for example, "Here, various authors have related the importance of family factors..."

I believe this is not an appropriate literature review in the present form. 

3) The presentation of results is some superficial. "With this work, it will be possible to maintain this type of centre from the point of 298
view of financial sustainability and quality, which will make it possible to maintain 299
the advantages that different authors have referred to this type of centre."

again different authors... this model... It would be much more appreciated if you use simple, but very direct words, describe your model in an easy to follow way and indicate who said and what in particular about it... 

4) There is a Spanish sentence in your paper, very odd... 304-307 lines, please remove it

Tabla 7 has the same problem... please do check your paper before submission!

5) The objective of the paper is still a big question mark for me. Authors say: "the main objective will be to develop a strategic plan 380
to improve the situation of this type of educational center - based on the analysis of 381
the proposed case - and to eliminate the economic problems derived from financial 382
and operational leverage"

But what is the theoretical contribution? What is the scientific novelty? How will this research bring forward our current knowledge. At present form, it is restricted to the best practice that authors suggest for educators to follow. But it is unclear, what the scientific community learns from it? What educators learn about the "transformation"? 

Even though you try to emphasize this is neither a plan nor a financial analysis, without elaboration of the above-mentioned questions, it still gives the impression of consulting work, not a scientific paper. 

All considered I feel that the contribution that the current version of this paper makes needs to be substantially improved. Much more focus needs to be given to the specific research question, which then needs to be carefully addressed in the literature review and the discussion of the results. 

Overall, I believe there is merit in the paper, but the revision must be substantial. I hope the comments provided in this review are helpful in further developing the manuscript. I wish you good luck in further developing this paper. 

Author Response

Esteemed reviewer,

thank you very much for the contributions to improve the work. They have been carefully taken into account and, we hope, we have been able to respond to them.

  • A new chapter on the study of global educational models has been introduced (lines 123-252). In this part, an attempt has been made to provide precise academic references on the different education systems. We have started from a global approach, then focused on the case of Spain.
  • In this part we have tried to improve the literature review used in the work. Seventeen references have been introduced (only in this section). An attempt has been made to link the assertions made in the paper to the specific authors (all this is also included in the "References" section). 
  • In the same way, a specific chapter  "Discussion" has been introduced in which we have tried to analyse from an academic point of view the positions for and against these co-educational models (lines 1017-1122).In this section we have also improved the overall literature review (twenty references). Finally, an attempt has been made to relate this review to the objectives and conclusions of the work.
  • Efforts have been made to improve the "Results" section. In this way, a series of initial considerations have been introduced on the results of the work (lines 723-743), linking them to the methodology developed and the conclusions obtained. Similarly, a final section referring to "Global Results" has been introduced (lines 964-1014). In this part, the aim is to assess a series of general considerations beyond the financial sphere. We believe that this part can help to improve the global dimension of the work.
  • The parts of the work that had not been translated into English have been corrected. Sincere apologies for this mistake (table 7, lines 304-307)
  • All of the above improvements have also been included in the "Conclusions" section. In this section, we have tried to reflect the theoretical contribution of the work (lines 1126-1145). Likewise, we have tried to include the most relevant contributions that the work could have from an academic point of view (1223 -1241) in order to extend its conclusions beyond the financial sphere.
  • Finally, an attempt has been made to improve the introduction of the section "Methodology" (lines 357-374) in order to link this section more closely to the overall structure of the work.

All changes are highlighted in blue to make them easier to locate.

Again, thank you very much for the contributions to improve the work. We hope to have the opportunity to keep on improving it. 

Best regards
